# Hall Effect Anisotropy in the Paramagnetic Phase of Ho_0.8_Lu_0.2_B_12_ Induced by Dynamic Charge Stripes

**DOI:** 10.3390/molecules28020676

**Published:** 2023-01-09

**Authors:** Artem L. Khoroshilov, Kirill M. Krasikov, Andrey N. Azarevich, Alexey V. Bogach, Vladimir V. Glushkov, Vladimir N. Krasnorussky, Valery V. Voronov, Natalya Y. Shitsevalova, Volodymyr B. Filipov, Slavomir Gabáni, Karol Flachbart, Nikolay E. Sluchanko

**Affiliations:** 1Prokhorov General Physics Institute of the Russian Academy of Sciences, 38, Vavilov Str., 119991 Moscow, Russia; 2Moscow Institute of Physics and Technology (State University), 141700 Moscow, Russia; 3Vereshchagin Institute for High Pressure Physics of RAS, 14 Kaluzhskoe Shosse, 142190 Troitsk, Russia; 4Institute for Problems of Materials Science, NASU, Krzhizhanovsky Str., 3, 03142 Kyiv, Ukraine; 5Institute of Experimental Physics SAS, 47, Watsonova, 04001 Košice, Slovakia

**Keywords:** dynamic charge stripes, anomalous Hall effect, Jahn-Teller instability, 72.15.Gd, 72.20.My

## Abstract

A detailed study of charge transport in the paramagnetic phase of the cage-cluster dodecaboride Ho_0.8_Lu_0.2_B_12_ with an instability both of the *fcc* lattice (cooperative Jahn–Teller effect) and the electronic structure (dynamic charge stripes) was carried out at temperatures 1.9–300 K in magnetic fields up to 80 kOe. Four mono-domain single crystals of Ho_0.8_Lu_0.2_B_12_ samples with different crystal axis orientation were investigated in order to establish the singularities of Hall effect, which develop due to (i) the electronic phase separation (stripes) and (ii) formation of the disordered cage-glass state below T*~60 K. It was demonstrated that a considerable intrinsic anisotropic positive component ρ^an^_xy_ appears at low temperatures in addition to the ordinary negative Hall resistivity contribution in magnetic fields above 40 kOe applied along the [001] and [110] axes. A relation between anomalous components of the resistivity tensor ρ^an^_xy_~ρ^an^_xx_^1.7^ was found for **H**||[001] below T*~60 K, and a power law ρ^an^_xy_~ρ^an^_xx_^0.83^ for the orientation **H**||[110] at temperatures T < T_S_~15 K. It is argued that below characteristic temperature T_S_~15 K the anomalous odd ρ^an^_xy_(T) and even ρ^an^_xx_(T) parts of the resistivity tensor may be interpreted in terms of formation of long chains in the filamentary structure of fluctuating charges (stripes). We assume that these ρ^an^_xy_(**H**||[001]) and ρ^an^_xy_(**H**||[110]) components represent the intrinsic (Berry phase contribution) and extrinsic (skew scattering) mechanism, respectively. Apart from them, an additional ferromagnetic contribution to both isotropic and anisotropic components in the Hall signal was registered and attributed to the effect of magnetic polarization of 5*d* states (ferromagnetic nano-domains) in the conduction band of Ho_0.8_Lu_0.2_B_12_.

## 1. Introduction

Numerous fundamental studies on strongly correlated electron systems (SCES) such as manganites [1,2,3,4], high-temperature superconducting (HTSC) cuprates [5,6,7,8], iron-based superconductors [9,10,11,12,13], chalcogenides [14], etc., have allowed for the discovery of a diversity of physical phenomena universal to SCES. Indeed, all these systems are characterized by a complexity of phase diagrams induced by strong phase separation due to structural or electronic instability [15]. The spatial electronic/magnetic inhomogeneity turns out to be directly related to simultaneously active spin, charge, orbital, and lattice degrees of freedom, which are considered as factors responsible for the appearance of high-temperature superconductivity in cuprates, as well as for the emergence of colossal magnetoresistance in manganites [5,16,17,18]. In particular, there are two possible mechanisms of the formation of spatially inhomogeneous ground states in SCES [19]: (i) disorder resulting from phase separation near a first-order metal–insulator transition caused by an external factor [19,20] and (ii) frozen disorder in the glass phase with short-range order formed by nanoscale clusters [21,22,23]. In the second case, one more and among the most significant mechanisms leading to an inhomogeneous glass state in HTSC oxides is the formation of static and dynamic charge stripes [24]. Such structures have been repeatedly observed in HTSC cuprates and nickelates by both direct and indirect techniques [25,26,27,28,29]. 

Studying the effect of spatial charge inhomogeneity on the scattering of charge carriers in HTSC cuprates, manganites, and other SCES is rather difficult due to their complex composition, low symmetry crystal structure, and high sensitivity to external conditions (pressure, magnetic field, etc., see, e.g., [4]). In this respect, it looks promising to use another model SCES—rare-earth dodecaborides (RB_12_). The RB_12_ (R—Tb, Dy, Ho, Er, Tm, Yb and Lu) attract considerable attention due to the unique combination of their physical properties, such as high melting point, microhardness, and high chemical resistance, which create prospects for practical applications [30,31,32,33]. These materials are also extremely interesting for fundamental studies. Indeed, both electronic (dynamic charge stripes) and structural (cooperative dynamic Jahn–Teller (JT) effect of the boron sub-lattice) instabilities take place in these high-boron borides with a simple *fcc* lattice (space group $Fm\bar{3}m$, see Figure 1a), in which stoichiometry can be reliably controlled during crystal growth [34].

Let us name the main factors that determine the appearance of spatial inhomogeneity, leading to symmetry lowering in *fcc* rare earth (RE) dodecaborides. Firstly, the cooperative dynamic Jahn–Teller effect in the rigid boron sub-lattice with covalent bonds that leads to lifting of degeneracy of the highest occupied molecular orbitals (HOMO) in B_12_ octahedrons and produces static structural distortions with related splitting E_JT_~500–1500 K (~50–150 meV) of HOMO [36]. Secondly, reaching the Ioffe–Regel limit near T_E_~150 K (~15 meV) causes a development of vibrational instability, which leads to an increase in the density of phonon states at T~T_E_ [37]. Thirdly, order–disorder transition to the cage-glass state at T*~60 K (~5–6 meV) causes random displacements of RE ions from central symmetric positions in B_24_ cuboctahedra, which form a rigid covalent boron framework [37]. Fourthly, well below T* large amplitude vibrations of neighbored RE ions in trigonal planes (transverse to the axis of the ferrodistortive JT effect [36]) produce periodic changes of hybridization between the 5*d*(RE) and 2*p*(B) states in the conduction band. This leads to the emergence of high-frequency charge density fluctuations with frequencies ν_S_~240 GHz [38] (denoted also as dynamic charge stripes, see Figure 1a) along one of the <110> direction in the *fcc* structure [39,40,41,42]. Stripe patterns are formed at characteristic temperature T_S_~*h*ν_S_/*k*_B_~15 K (~1 meV). These structural and electronic instabilities initiate nanoscale phase separation and inevitably cause strong charge transport anisotropy in external magnetic field both in the nonmagnetic reference LuB_12_ [37] and in magnetic RB_12_ [41,42,43,44,45,46]. In particular, the above features of the crystal and electronic structure of RB_12_ have a decisive effect on the characteristics of charge transport when Lu ions (with a filled *f*-shell, 4f^14^ configuration) are partially replaced by magnetic Ho (4f^10^) ions in Ho_x_Lu_1-x_B_12_ compounds.

The spatial inhomogeneity of fluctuating electron density is the origin for the strong anisotropy of magnetic phase diagrams in these systems (see, e.g., Figure 1b–d for the best quality single-domain crystals of Ho_0.8_Lu_0.2_B_12_ and [35,43,44,45,46]). Indeed, strong magnetic anisotropy is observed, for instance, in Ho_x_Lu_1-x_B_12_ with a high concentration of magnetic ions both in the paramagnetic (P) state (see the color plot in Figure 1e demonstrating the anisotropy of magnetoresistance) and on the angular antiferromagnetic (AF) phase diagrams, which reveal a Maltese cross-symmetry (see Figure 1f,g, [35,43,44] and also [46] for TmB_12_). It is worth noting that, like in the nonmagnetic reference compound LuB_12_, strong charge transport anisotropy is observed in the paramagnetic state of Ho_x_Lu_1-x_B_12_, ErB_12_ [45], and TmB_12_ [46] (see, for example, Figure 1e and [47]) and attributed to interaction of electron density fluctuations (stripes) with external steady magnetic field (for recent review see [48] and references therein). 

Until now, studies of electron transport in Ho_x_Lu_1-x_B_12_ have mainly focused on transverse magnetoresistance (see, e.g., [35,43,47]). Nevertheless, the recent study of LuB_12_ [49] and the initial short research on Ho_0.8_Lu_0.2_B_12_ (Ref. [50]) have demonstrated a significant anisotropy of the Hall effect due to an anomalous positive anisotropic contribution that appeared below T*~60 K. Thus, it is of great interest to study in detail the effect of electronic phase separation on the off-diagonal component of the resistivity tensor in model magnetic compound Ho_0.8_Lu_0.2_B_12_ with dynamic charge stripes. As a continuation of the short study conducted in [50], this work presents results and detailed analyses of the normal and anomalous contributions to the Hall effect in the paramagnetic phase of Ho_0.8_Lu_0.2_B_12_. We investigated both the angular and magnetic field dependences of Hall resistivity in detail and determined the anisotropic component of the resistivity tensor for this model system with electronic phase separation (dynamic charge stripes). The observed complex angular behavior of the anisotropic Hall resistivity is attributed to interaction of the filamentary structure of fluctuating charges with the external magnetic field. The arguments presented here favor both intrinsic and extrinsic mechanisms of the anomalous Hall effect formation.

## 2. Experimental Results and Data Analysis

### 2.1. Temperature Dependences of Resistivity and Hall Resistivity

In conventional Hall effect experiments the Hall coefficient is calculated as R_H_ = ρ_H_/H = ((V_H_(+H) − V_H_(–H))/(2I))·d/H, where d is the sample thickness, I the excitation current and V_H_(±H) the voltage measured on Hall probes in two opposite directions of external magnetic field. Taking into account the complex field dependence of Hall effect in parent compound LuB_12_ [49], and the different origin of the detected anomalous contributions to Hall resistivity [51], the term “reduced Hall resistivity” for ρ_H_/H is used below in the present study instead of the Hall coefficient R_H_. 

Figure 2 shows the temperature dependences of resistivity ρ(T) at H = 0 and 80 kOe, as well as the reduced Hall resistivity ρ_H_(T)/H in Ho_0.8_Lu_0.2_B_12_ calculated from experimental results recorded for three different crystals with principal directions **H**||**n**||[001], **H**||**n**||[110], and **H**||**n**||[111], and identical DC current direction **I** || [$\bar{110}$]. Vertical solid lines point to the transition to the cage-glass state at T*~60 K [37] and to the formation of stripes at T_S_~15 K (see also discussion below). In the zero-magnetic field, the ρ(T) curves measured for all three Ho_0.8_Lu_0.2_B_12_ samples correspond to metallic conductivity with the same RRR value ρ(300 K)/ρ(4.2 K) = 13.47 (Figure 2a). The data for H = 0 kOe and 80 kOe are clearly separated below T*~ 60 K, indicating a pronounced sign-alternating magnetoresistance. Note that the ρ(T, H = 80 kOe) curves for crystals with **n**||[110] and **n**||[111] match together above the characteristic temperature T_S_~15 K and differ noticeably at lower temperatures. On the contrary, at T < T*~60 K, the ρ(T, H = 80 kOe) dependence for field direction **H**||**n**||[001] lies well above those for **H**||**n**||[110] and **H**||**n**||[111], so the MR anisotropy reaches values ρ(**n**||[001])/ρ(**n**||[111]) ≈ 1.8 (MR~80%) at T = 2.1 K. Open symbols in Figure 2b show the results of ρ_H_/H(T) measurements in the scheme with two opposite orientations of **H**. Significant differences between the ρ_H_/H(T) dependences for different field directions appear below T*~60 K, while the curves for samples with **H**||**n**||[110] and **H**||**n**||[111] start to diverge below 15 K. In this case, the lowest negative values of ρ_H_/H(T) are detected for the **n**||[001] sample, while the highest values are observed for **H**||**n**||[111]. The maximal anisotropy of the reduced Hall resistivity at T = 2.1 K and H = 80 kOe equals to ρ_H_/H(**n**||[111])/ρ_H_/H(**n**||[001]) ≈ 1.8 (~80%), which is very similar to resistivity anisotropy. Thus, the temperature dependences of ρ_H_/H(T) allow to identify some anisotropic positive component of the Hall signal, which appears in Ho_0.8_Lu_0.2_B_12_ in strong magnetic fields. It is worth noting that temperature-lowering results in the increase in anisotropy for both Hall resistivity and MR components (Figure 2).

### 2.2. Field Dependences of Hall Resistivity and Magnetization

Figure 3a–c show the reduced Hall resistivity ρ_H_/B(B) vs magnetic induction **B** at temperatures of 2.1, 4.2, and 6.5 K measured in the conventional scheme on three different samples with magnetic field **H** applied along their normal vectors- **n**||[001], **n**||[110], and **n**||[111], correspondingly. The related magnetic susceptibility M/B(B) is shown in Figure 3d–f and Figure 4 shows the temperature dependence of magnetic susceptibility M/B(T) ≡ χ(T) measured in magnetic field H = 100 Oe. The data were corrected by demagnetizing fields. It is seen from Figure 3d–f, that M/B(B) decreases with increasing both field and temperature in the paramagnetic phase, indicating a trend towards saturation of magnetization in strong magnetic fields. It can be discerned from Figure 3e,f that in the paramagnetic state, these M/B(B) curves are very similar, and the magnetic anisotropy at H~70 kOe does not exceed 1.4% even at lowest available temperature 2.1 K. Therefore, below we analyze the Hall effect using the same dependence M/B(**n**||[001]) for all three orientations of applied magnetic field.

The AF-P phase transition at *T*_N_ = 5.75 K can be clearly recognized on the temperature dependence of magnetic susceptibility measured at H = 100 Oe (see Figure 4). Above *T*_N_, the low field magnetic susceptibility χ(*T*) may be described approximately by a Curie–Weiss type dependence
(1)$$\chi =M/H=N_{\mathrm{Ho}}\;{\mu_{\mathrm{eff}}}^{2}/(3kB_{\;}(T-\theta_{p}))+\chi_{0}$$
where *N*_Ho_ = 0.95·*x*(Ho)·10^22^ cm^−3^, and *μ*_eff_~10 *µ*_B_ are the concentration and the effective magnetic moment of Ho-ions, correspondingly (*µ*_B_ and *k*_B_ denote Bohr magneton and Boltzmann constant), *θ*_p_ ≈ −14 K is the paramagnetic Curie temperature corresponding to AF exchange between magnetic dipoles. *χ*_0_ ≈ −1.78·10^−3^ *µ*_B_/mole/Oe is the temperature-independent additive combination of (i) diamagnetic susceptibility of the boron cage and (ii) Pauli paramagnetism and Landau diamagnetism of conduction electrons. 

Fitting of *χ*(T) by Equation (1) with temperature-dependent *μ*_eff_(T) indicates that within experimental accuracy the susceptibility follows the Curie–Weiss dependence with magnetic moment, which is only slightly below the total moment *μ*_eff_ ≈ 10.6 *µ*_B_ of Ho^3+^ 4*f*-shell in the range 80–300 K. As the population of excited magnetic states of the Ho^3+ 5^*I*_8_ multiplet (that is split by crystalline electric field (CEF) [52]) declines significantly in the range 8–80 K, *μ*_eff_ decreases moderately (to 9.5 *µ*_B_; see Figure 4, right scale). Thus, even at *T*_N_, the value of *μ*_eff_ noticeably exceeds the magnetic moment of the Γ_51_ ground state triplet *μ*_eff_ (Γ_51_) ≈ 8 µ_B_ (solid line in Figure 4, right scale). The difference (∆*μ*_eff_~1.5 *µ*_B_, Figure 4) may be related to ferromagnetic correlations, which develop in this SCES below *T**~60 K. Note that below 25 K, various short-range ordering features including ferromagnetic components were previously observed in magnetic RB_12_ [53,54,55].

As can be seen from Figure 3a–c, the behavior of reduced Hall resistivity ρ_H_/B(B) differs significantly depending on **B** direction. Indeed, in the paramagnetic region for **B**||**n**||[001] the value of ρ_H_/B(B) turns out to decrease, for **B**||**n**||[110] the curve is practically field independent, and for **B**||**n**||[111], an increase of negative ρ_H_/B(B) values is observed. These trends persist in temperature range 2.1–6.5 K in the paramagnetic phase (above Neel field, B > B_N_ in Figure 3a–c), and the anisotropy of ρ_H_/B(**n**||[001])/ρ_H_/B(**n**||[111]) reaches values of ~80% at 2.1 K for B = 80 kG in accordance with the results of Figure 2b. Such strong anisotropy is very unusual for the paramagnetic state of *fcc* metals (as Ho_x_Lu_1-x_B_12_) with intense charge carrier scattering in the disordered cage-glass phase.

### 2.3. Angular Dependences of Hall Resistivity in the Paramagnetic State of Ho_0.8_Lu_0.2_B_12_

To reveal the nature of the strong anisotropy of ρ_H_/H Hall resistivity (Figure 2b and Figure 3a–c) as well as to separate different contributions to Hall effect, it is of interest to study the angular dependencies of Hall resistivity ρ_H_(φ) in Ho_0.8_Lu_0.2_B_12_ for different configurations of magnetic field with respect to principal crystallographic directions. Here we present precision measurements of Hall resistivity ρ_H_(φ,T_0_,H_0_,**n**) angular dependencies performed at 2.1–300 K in magnetic field up to 80 kOe for four crystals of Ho_0.8_Lu_0.2_B_12_ with different orientations of normal vector **n** to sample surface: **n**||[001], **n**||[110], **n**||[111], and **n**||[112] (see inset in Figure 2a). In these cases, each sample was rotated around current axis **I**||[1,2,3,4,5,6,7,8,9,10]. Thus, both fixed vector **H** and rotating vector **n** were lying in the same plane (1–10). For clarity, Appendix A demonstrates a direct correlation between the results of two different measurements of the Hall effect: (i) in the conventional field-sweep scheme with two opposite directions of ±**H**||**n** and (ii) in the step-by-step rotation of the sample around **I**||[1,2,3,4,5,6,7,8,9,10] with a fixed **H** vector in the plane perpendicular to the rotation axis (see the inset in Figure 2a).

Figure 5 shows the results of angular ρ_H_(φ) measurements at H = 80 kOe for samples with normal directions **n**||[001] and **n**||[110] in temperature ranges 40–300 K (Figure 5a,c) and 2.1–25 K (Figure 5b,d). The experimental results were fitted by formula.

ρ_H_(φ) = ρ_H_^const^ + ρ_H0_·cos(φ + φ_sh_) + ρ_H_^an^(φ)(2)
where ρ_H_^const^ is an angle independent component, ρ_H0_ is the amplitude of the isotropic cosine-like contribution to Hall resistivity f_cos_(φ) = ρ_H0_·cos(φ + φ_sh_), φ_sh_ is the phase shift, and ρ_H_^an^(φ) = ρ_H0_^an^·g(φ) the anisotropic contribution to Hall resistivity (see Figure 5). The approximation of ρ_H_(φ) within the framework of Equation (2) for two crystals with normal directions **n**||[001] and **n**||[110] was carried out in two intervals Δφ = 90 ± 35° and Δφ = 270 ± 35° where the cosine-type behavior is almost perfect. By analogy, ρ_H_(φ) curves for samples with **n**||[111] and **n**||[112] were approximated in same intervals Δφ = 90 ± 35° and Δφ = 270 ± 35° (near the zeros of angular dependencies), but without reference to certain crystallographic directions (see Appendix A). As a result, the ρ_H0_(T, H, **n**) and φ_sh_(T, H, **n**) parameters of the isotropic contribution f_cos_(φ) in (2) were found directly from this approximation. The anisotropic contribution ρ_H_^an^(T, H, **n**) at fixed direction **H**||**n** was determined as an average of the sum of absolute ρ_H_^an^(φ) values found for **n** at φ = 0°, φ = 180°, and φ = 360° in the rotation experiment. As can be seen from the analysis of angular ρ_H_^an^(φ) dependencies undertaken below, the proposed approach reveals significant limitations and inaccuracies inherent in Hall effect measurements according to the conventional field-sweep scheme. Taking into account that ρ_H_^const^ and φ_sh_ ≈ 3–5° lead only to small corrections in determining the ρ_H0_ = ρ_H0_(T, H, **n**) and ρ_H_^an^ = ρ_H_^an^(T, H, **n**) amplitudes in Equation (2), the experimentally measured Hall resistivity is discussed below as a sum of isotropic and anisotropic contributions ρ_H_(φ) ≈ ρ_H0_·cos(φ) + ρ_H_^an^ cos(φ)·g(φ).

In the range 40–300 K at H = 80 kOe the experimental data for **n**||[001] and **n**||[110] samples (Figure 5a,c) are well fitted by a cosine dependence, indicating the absence of anisotropic contribution—ρ_H_^an^(φ)~0. On the contrary, below 40 K, the ρ_H_^an^(φ) curves for **n**||[001] exhibit a broad feature in a wide range of angles around <001> (between nearest <111> axes (see Figure 5b)) with a step-like singularity just at <001>. Several peaks of relatively small amplitude may be identified on ρ_H_^an^(φ) dependence for the **n**||[110] sample (Figure 5d). The ρ_H_^an^(φ) curves for samples with **n**||[111] and **n**||[112] in the range 2.1–30 K and at H = 80 kOe are presented in Appendix A (see Appendix A). Note that the ρ_H_(φ) dependences for **n**||[111] and **n**||[112] being similar to each other differ from curves recorded for **n**||[001] and **n**||[110] samples, and deviate significantly from cosine dependence in a wide range of angles. The anisotropic contribution of ρ_H_^an^(φ) extracted for **n**||[111] and **n**||[112] samples is close to zero near their normal directions **n** (for more details see Appendix A).

Figure 6 shows the result of approximation by Equation (2) of the measured Hall resistivity ρ_H_(φ) at T = 6.5 K in fixed fields up to 80 kOe for the studied four crystals. It is seen that the anisotropic contribution ρ_H_^an^(φ) appears just above 40 kOe having the largest amplitude for sample **n**||[001], it decreases by a factor of 2 for **n**||[110] and goes to zero for **n**||[111] and **n**||[112] samples (Figure 6). Indeed, below 40 kOe the experimental data (shown by symbols) and the cosine fits (thin solid lines) coincide with a good accuracy indicating the absence of an anisotropic component ρ_H_^an^(φ) in the low field region (see also Appendix A for the **n**||[111] at T = 20 K).

It is worth noting that in the range T > T*~60 K, the temperature dependences of reduced Hall resistivity ρ_H_/H(T) at H = 80 kOe for samples with **n**||[001], **n**||[110], and **n**||[111] coincide within the experimental accuracy (Figure 2b). Angular ρ_H_(φ) curves can be fitted well by cosine (Figure 5a,c), and they are close to each other. Note also that for **n**||[110] and **n**||[111] samples, the amplitudes ρ_H0_ of ρ_H_(φ) coincide in an even wider temperature range of 15–60 K (Figure 2b). Below T_S_~15 K and in the range H > 40 kOe, a significant deviation of the ρ_H_/H(T) curves from cosine-type behavior is observed for crystals with **n**||[110], **n**||[111], and **n**||[112] in intervals Δφ = 90 ± 35° and Δφ = 270 ± 35° (see, e.g., Figure 6). This can lead to large errors in determining the amplitude ρ_H0_ of the main contribution from conventional ±**H** field-sweep measurements. In our opinion, this finding allows us to explain the different behavior of reduced Hall resistivity ρ_H_(T)/H for various field directions (see Figure 2) and shed light on the shortcoming of conventional ±**H** field-sweep scheme commonly used for the Hall effect studies. Thus, avoiding incorrect conclusions, at low temperatures and in magnetic fields H > 40 kOe, an isotropic ordinary contribution to the Hall effect is assumed, and one common ρ_H0_ value found from the analysis of Equation (2) for **n**||[001], which was used for these four studied crystals with different **n** directions. At the same time, in fields H ≤ 40 kOe at T < T_S_~15 K, the ρ_H_/H(T) curves in intervals φ = 90 ± 35° and φ = 270 ± 35° differ only slightly from cosine curves (Figure 6); therefore, approximation by Equation (2) was carried out with individual parameters of the harmonic contribution for each of the four samples.

### 2.4. Analysis of Contributions to Hall Resistivity

Figure 7 shows the fitting parameters obtained from the approximation by Equation (2) (Figure 5 and Figure 6 and Appendix A) of ρ_H_(φ) dependences in the paramagnetic (P) phase of Ho_0.8_Lu_0.2_B_12_. Different symbols correspond to isotropic ρ_H0_/H(T_0_,H) and anisotropic ρ_H_^an^/H(T_0_,H) components estimated at T_0_ = 2.1, 4.2, and 6.5 K (vertical dashed lines in Figure 2 denote the T_0_ values). The data for different samples with **n**||[001], **n**||[110], **n**||[111], and **n**||[112] in Figure 7 are indicated by different colors. It is seen in Figure 7a that at T_0_ = 6.5 K for the sample with **n**||[001], the value of ρ_H0_/H is about field independent below 40 kOe, while in the range H > 40 kOe, the negative values of ρ_H0_/H(T_0_,H) increase linearly. For **n**||[110], **n**||[111], and **n**||[112] samples, the negative values of ρ_H0_/H(T_0_,H) decrease moderately with increasing magnetic field below 40 kOe. Note that in P-phase, the variation of the isotropic ρ_H0_/H(T_0_,H) component may be attributed to significant (~14%) and non-monotonous change of the concentration of conduction electrons if we assume one type of charge carrier with relation ρ_H0_/H = R_H_(T)~1/*n*_e_*e* (*e* is the electron charge, and *n*_e_ the concentration of charge carriers). For convenience, the reduced Hall concentration *n*_e_/*n*_R_ = (H/ρ_H0_)/*en*_R_ is shown on right axis of Figure 7a, where *n*_R_ = 0.95*10^22^ cm^−3^ is the concentration of Ho and Lu atoms. Note also that the amplitude of anisotropic contribution ρ_H_^an^/H(T_0_,H) turns out to be almost zero below 40 kOe. In a stronger magnetic field H > 40 kOe, the anisotropic component increases for samples with **n**||[001] and **n**||[110] (Figure 7b), with the amplitude ρ_H_^an^/H(H) for **n**||[001] being more than two times higher than the corresponding values for **n**||[110]. For **n**||[111] and **n**||[112] directions, the small negative values turn out to be close to zero (see also Appendix A). 

The experimental temperature dependences ρ_H_/H(T,H_0_ = 80 kOe) obtained in the conventional, commonly used the field-sweep technique with two opposite orientations of applied magnetic field ±**H**||**n** from one side, and the isotropic component deduced from the angular dependences of Hall resistivity ρ_H0_/H(T,H_0_) from the other, are compared in Figure 2b. It can be seen that for the sample with **n**||[111], the ρ_H0_/H(T,H_0_) data coincide with good accuracy with the ρ_H_/H values detected by conventional field-sweep measurements in a wide range of temperatures 1.9–300 K. For the sample with **n**||[110], ρ_H0_/H starts to deviate from ρ_H_/H(T,H_0_) at T < T_S_~15 K, and for **n**||[001], noticeable differences arise already upon the transition to the cage-glass state at T*~60 K (Figure 2b). This observation allows to attribute the appearance below T* of the strong anisotropy of the Hall effect in Ho_0.8_Lu_0.2_B_12_ to the contribution ρ_H_^an^/H, which increases additionally below T_S_. Figure 2b shows a comparison of the parameter sum = ρ_H0_/H + ρ_H_^an^/H, which corresponds to Hall effect amplitude detected from ρ_H_/H angular dependences from one side, and the reduced Hall resistivity measured in the conventional ±**H**||**n** field-sweep experiment from the other. It can be seen that the temperature behavior of the sum detected by angular measurements is in good agreement with the results of conventional ±**H**||**n** field-sweep dependences of ρ_H_/H(H) for all crystals under investigation (Figure 2b, **H**||**n**||[001], **H**||**n**||[110], and **H**||**n**||[111]) approving the validity of the proposed Hall effect separation. Then, using the simple relation ρ_H0_/H(T) = R_H_(T)~1/*n*_e_*e*, we roughly estimate from the temperature dependences presented in Figure 2b and the charge carrier concentration *n*_e_ in the conduction band. Figure 8a shows the strong field (H = 80 kOe) Arrhenius plot *lg*(*n*_e_/*n*_R_)~1/*T* for ***H****||***n**||[001] and ***H****||***n**||[110] directions, similar to Figure 7a (right axis), demonstrating the field dependence of the ratio *n*_e_/*n*_R_.

It can be seen in Figure 8a that the Arrhenius-type dependence 1/R_H_(T)~*e·n*_0_·exp(−T_a_/T) is valid above the cage-glass transition at T*~60 K, and that the estimated activation temperatures T_a_ = 14.4 ± 0.6 K and T_a_ = 16.7 ± 0.3 K detected for **H**||**n**||[001] and **H**||**n**||[110], correspondingly, are close to T_S_~15 K. As the detected initial concentration *n*_0_ = (1.61–1.64)·10^22^ cm^−3^ coincides within experimental accuracy for these two field directions, the reduced Hall concentration changes in the range *n*_e_/*n*_R_ = 1.2–1.6 (Figure 8a) in accordance with the results of previous Hall effect measurements in RB_12_ (R = Lu, Tm, Ho, Er) [56].

Figure 8b shows the temperature dependences of Hall mobility *µ*_H_(T) ≈ ρ_H0_(T)/[H·ρ(T)] (left scale) and the related parameter *ω*_c_*τ* ≈ *µ*_H_·*H* (right scale, where *ω*_c_ is the cyclotron frequency and *τ* the carrier relaxation time) in magnetic field H = 80 kOe for samples with **n**||[001], **n**||[110], and **n**||[111]. At low temperatures T < T_S_~15 K the obtained *µ*_H_(T) data tend to constant values *µ*_H_~600–700 cm^2^/(V·s) (Figure 8b), and in the range T > T*~60 K Hall mobility follows the power law *µ*_H_~*T*^−*α*^ with a single exponent being estimated as *α* ≈ 5/4 both for **H**||**n**||[001] and **H**||**n**||[110]. 

Similar behavior of Hall mobility was observed in the range 80–300 K previously for various LuB_12_ crystals; an *α* ≈ 7/4 exponent was detected for crystals with large values of RRR ≡ ρ(300 K)/ρ(6 K) = 40–70 and *α* ≈ 3/2 was estimated for LuB_12_ with a small RRR~12 [49]. The *α* = 3/2 exponent is typical for scattering of conduction electrons by acoustic phonons (deformation potential). The increase of *α* values up to 7/4 in best LuB_12_ crystals was interpreted [49] in terms of charge carriers scattering both by quasi-local vibrations of RE ions and by boron optical phonons [57] in the presence of JT distortions and rattling modes of RE ions [58,59,60]. In the case of Ho_0.8_Lu_0.2_B_12_ crystals with RRR~10 the further decrease of *α* value from 3/2 to 5/4 could be attributed to the emergence of strong magnetic scattering in dodecaboride with Ho^3+^ magnetic ions. Note that the inequality *ω_c_τ* < 1 (see Figure 8b), which is fulfilled in the entire temperature range 1.9–300 K in fields up to 80 kOe, corresponds to the low field regime of charge carriers in Ho_0.8_Lu_0.2_B_12_. This indicates that the results of Hall effect measurements should be insensitive to the topology of the Fermi surface and depending mainly on the features of disorder and inhomogeneity of the crystals studied.

Obviously, the Hall effect in Ho_0.8_Lu_0.2_B_12_ is strongly modified by the positive anisotropic contribution ρ_H_^an^/H. Figure 9a,b show the angular dependencies of ρ_H_^an^(φ)/H at H = 80 kOe for samples with **n**||[001] and **n**||[110]. The same contributions for **n**||[111] and **n**||[112] are shown in Figure 10. As can be seen from Figure 9 and Figure 10, the ρ_H_^an^(φ)/H curves differ in both the amplitude and shape of angular dependence. Since very small changes of ρ_H_^an^(φ)/H with temperature are detected for samples with **H**||**n**||[111] and **H**||**n**||[112] in the field along normal directions (φ = 180° in Figure 10), the temperature dependences of the anomalous component below are analyzed only for **n**||[001] and **n**||[110].

The amplitude ρ_H_^an^/H variation with temperature is shown in Figure 9c,d in the logarithmic plot. It is worth noting that two phenomenological relations

ρ_H_^an^/H ≈ C*·(1/T − 1/T_E_)(3)ρ_H_^an^/H ≈ (ρ_H_^an^/H)_0_ −A_H_·T^−1^·exp(−T_a_^H^/T)(4a)
were used in [49,55,61] for the Hall effect analysis in strongly correlated electron systems with a filamentary structure of conducting channels. Among these, a hyperbolic dependence (3) of Hall resistivity was observed previously in SCES CeCu_6-x_Au_x_ [55] and Tm_1-x_Yb_x_B_12_ [61]. The authors of [55,61] argued in favor of a transverse even component of the Hall signal associated with the formation of stripes (see also [38]) on the surface and in the bulk of the crystal, similar to the chains of nanoscale stripes detected in the normal phase of HTSC [62]. Equation (4a) was applied to discuss the temperature induced destruction of the coherent state of stripes in LuB_12_ [49]. Below, we use the analysis based on Equations (3) and (4a) to highlight quantitatively the changes caused by various orientations of the external magnetic field.

Indeed, the approximation by Equation (3) in the range 5–40 K results in values *T*_E_ ≈ 132 K and *C** ≈ 8.9·10^−4^ cm^3^/C for the sample with **n**||[001], while for **n**||[110], the parameter *T*_E_ ≈ 135 K is almost the same and *C** ≈ 4.3·10^−4^ cm^3^/C turns out to be about half the size (see green curves in Figure 9c,d). Then, the analysis based on Equation (4a) provides very similar values T_a_^H^ ≈ 13.7–13.8 K for these two field directions (see Figure 9c,d, red curves). Note that the estimation of T_a_^H^ agrees both with the characteristic temperature of stripe chains formation T_S_~15 K [48] from one side, and with the activation energy E_a_/k_B_ =T_a_~14–16 K detected above in the Arrhenius type approximation of the main contribution to the Hall effect (see Figure 8a) from the other. It can be seen from Figure 9c,d that for the case **n**||[001] and **n**||[110], Equation (4a) provides a good description of the experimental ρ_H_^an^/H(T) curves at temperatures up to 10 K. Above 10 K, these fits (see red curves in Figure 9c,d) differ sharply from experiment, indicating the restriction of the phenomenological approach applied. The A_H_ coefficients in Equation (4a) differ for these two field directions by more than two times (A_H_ = 73.8·10^−4^ cm^3^/C for **n**||[001] and A_H_ = 32.3·10^−4^ cm^3^/C for **n**||[110]), being in good agreement with the amplitude ratio for ρ_H_^an^/H (Figure 9c,d). Furthermore, similar to the approach developed in [43] for LuB_12_, an analysis of the anisotropic positive contribution to magnetoresistance in Figure 9e,f for Ho_0.8_Lu_0.2_B_12_ is carried out within the relation

ρ_xx_^an^(**n**,T_0_,H = 80 kOe) ≈ (ρ_xx_^an^)_0_ − A_xx_·T^−1^·exp(−T_a_^ρ^/T)(4b)

For each of the samples with **H**||**n**||[001] and **H**||**n**||[110], the anisotropic component ρ_xx_^an^(**n**,T_0_,H = 80 kOe) was determined by subtracting from the experimental resistivity data (e.g, ρ(**n**||[001],T_0_,H = 80 kOe) for **H**||**n**||[001]) and the dependence ρ(**n**||[111],T_0_,H = 80 kOe) for **H**||**n**||[111], where the magnetoresistance is minimal. Parameters T_a_^ρ^ = 13.3–14.8 K found from this approximation in the same range T ≤ 10 K turn out to be close to T_a_^H^ ≈ 13.3–13.8 K values and also to T_S_~15 K [48], as well as to the activation temperature T_a_~14–16 K (see Figure 8a).

## 3. Discussion

### 3.1. Multicomponent Analysis of the Contributions to Anomalous Hal Effect (AHE) in the Regime of Ferromagnetic Fluctuations

Previous measurements of the Hall effect in Ho_0.5_Lu_0.5_B_12_ in P-phase (T > T_N_ ≈ 3.5 K) were carried out in the conventional field-sweep scheme with two opposite field directions ±**H**||**n** [63]. Ordinary and anomalous components of the Hall effect observed in [63] were described by the general relation

ρ_H_/B = R_H0_ + R_S_·4πM/B,(5)
which is usually applied to AHE in ferromagnetic metals [51,64], where R_H0_ and R_S_ = const(T) are the ordinary and anomalous Hall coefficients, respectively. According to [51], the ferromagnetic AHE regime represented by Equation (5) corresponds to the intrinsic scattering mechanism. Since short-range order effects are observed in the paramagnetic phase of magnetic RB_12_ in the temperature range at least up to 3*T*_N_ [53,60,65] (see also Figure 4), and that a ferromagnetic component was detected in the magnetically ordered state of HoB_12_ in magnetic fields above 20 kOe, it is of interest to perform the analysis within the framework of Equation (5) of the ordinary and AHE components in the vicinity of *T*_N_. In this case, relying on the above results of angular measurements of Ho_0.8_Lu_0.2_B_12_ (Figure 5, Figure 6, Figure 7, Figure 9 and Figure 10), one should use isotropic ρ_H0_(T, H, n) and anisotropic ρ_H_^an^(φ, T, H, n) components of the Hall signal separated within the framework of Equation (2) (see Figure 2, Figure 3, Figure 4, Figure 5, Figure 6, Figure 7, Figure 8, Figure 9 and Figure 10). It is worth noting that the analysis performed in [43] within the framework of Equation (5) is applied to the field dependences of Hall resistivity ρ_H_(H) obtained in the conventional field-sweep ±**H** || **n** scheme, leading obviously to mixing of contributions ρ_H0_(H) and ρ_H_^an^(H). As a result, the coefficients R_H0_ and R_S_ were determined in [43] for the total averaged Hall resistivity, which contains both the isotropic ρ_H0_(T, H, n) and anisotropic ρ_H_^an^(φ, T, H, n) components. Actually, each of the angular contributions ρ_H0_ and ρ_H_^an^(φ) is characterized by two independent coefficients R_0_ and R_S_, which generally differ in sign. Below, we develop the analysis, considering two ferromagnetic components included in the AHE. To keep generality, we analyze our Hall effect data for all three principal directions of external magnetic field (**H**||[001], **H**||[110], and **H**||[111]), despite the fact that for **H**||[111], the intrinsic AHE is found to be practically negligible (see Figure 7 and Figure 10).

Thus, for a full description of the Hall effect in Ho_0.8_Lu_0.2_B_12_, we used the following relations:(6)$$\left\{\begin{array}{c} \rho_{H}^{0}/B=R_{H}^{0}+R_{M}^{0}*4\pi M/B\\ \rho_{H}^{\mathrm{an}}/B=R_{H}^{\mathrm{an}}+R_{M}^{\mathrm{an}}*4\pi M/B \end{array}\right.$$
where ρ_H_^0^/B = ρ_H0_/B(H, T_0_, n) and ρ_H_^an^/B = ρ_H_^an^/B(H, T_0_, n) are the reduced amplitudes of contributions ρ_H0_ and ρ_H_^an^(φ) (see Figure 7), which depend on temperature and direction of the normal vector **n** to the sample surface; R_H_^0^ and R_H_^an^ are components of the ordinary Hall effect, connected with magnetic induction B, and R_M_^0^, R_M_^an^ are the coefficients of anomalous Hall effect determined by magnetization M and related to the ferromagnetic component (see Figure 3d–f). Obviously, there are two independent ordinary contributions to Hall resistivity R_H_^0^·B and R_H_^an^·B, as well as two independent anomalous (ferromagnetic) components R_M_^0^·4πM and R_M_^an^·4πM, which differ for various **H** directions. Note that the last two components R_H_^an^·B and R_M_^an^·4πM are responsible for the observed Hall effect anisotropy. 

Figure 11 shows the linear approximation within the framework of Equation (6) of the reduced amplitudes ρ_H0_/B and ρ_H_^an^/B *vs* 4πM/B in the range T_0_ = 2.1–6.5 K for Ho_0.8_Lu_0.2_B_12_ crystals with **n**||[001], **n**||[110], and **n**||[111]. Linear fits are acceptable for directions **n**||[001] and **n**||[110] at temperatures of 2.1 K and 4.2 K for both the isotropic ρ_H0_/B and anisotropic ρ_H_^an^/B contributions. This approximation was found to be valid at T = 6.5 K in the interval of small 4πM/B values (i.e., in high magnetic fields, in more detail see Appendix A), where the estimated parameters R_H_^0^ and R_H_^an^ are extracted as cutoffs, and R_M_^0^ and R_M_^an^ are the slopes of corresponding straight lines in Figure 11. Since the parameters R_H_^0^, R_H_^an^, R_M_^0^, R_M_^an^ depend weakly on temperature (see Appendix A), the temperature-averaged values of these coefficients are summarized in Table 1.

The analysis based on Equation (6) allows us to conclude that the values of coefficients R_H_^an^ and R_M_^an^, which are characteristics of the anisotropic component, turn out to be practically equal to zero for **H**||[111] (Figure 11f). It can be seen that the Hall coefficient R_H_^0^~−7.5·10^−4^ cm^3^/C remains practically invariant for any **H** direction (see Table 1 and Appendix A), confirming that this ordinary negative component of Hall signal is isotropic. On the contrary, the value of anisotropic positive contribution R_H_^an^ changes significantly from ~6.8·10^−4^ cm^3^/C for **H**||[001] to 2.3·10^−4^ cm^3^/C in **H**||[110] passing through zero for **H**||[111] (Table 1). As a result, in Ho_0.8_Lu_0.2_B_12_ for strong field **H**||[001], the anisotropic positive component R_H_^an^ B·g(φ,n), which is proportional to magnetic induction, turns out to be comparable in absolute value with the negative isotropic component R_H0_ B of the ordinary Hall effect. Note that the values of coefficients R_M_^0^, R_M_^an^ of the anomalous (ferromagnetic) contributions, which are proportional to magnetization, dramatically exceed the ordinary parameters R_H_^0^, R_H_^an^ that agrees with the result [63] for Ho_0.5_Lu_0.5_B_12_.

It is very unusual that a significant isotropic anomalous positive contribution R_M_^0^ ~25:27·10^−4^ cm^3^/C appears in the paramagnetic phase of Ho_0.8_Lu_0.2_B_12_, and it may be attributed to the isotropic ferromagnetic component in the Hall signal. We propose that the R_M_^0^ term depends on the regime of ferromagnetic fluctuations detected in low field magnetic susceptibility above T_N_ (Figure 4). On the contrary, the anomalous negative contribution R_M_^an^ varies strongly in the range (−)(21:65)·10^−4^ cm^3^/C depending on **M** direction (see Table 1), and this component appears in strong magnetic field and at low temperatures (Figure 11). To summarize, AHE in the paramagnetic state of Ho_0.8_Lu_0.2_B_12_ is proportional to magnetization and is determined both by the positive contribution R_M_^0^ ·4πM·cos(φ) and by the strongly anisotropic negative component R_M_^an^·4πM·g(φ). These two contributions compensate each other in the vicinity of **n**||[110] (dynamic charge stripe direction [35,48]). 

When discussing the nature of multicomponent Hall effect in Ho_0.8_Lu_0.2_B_12_, it is worth noting the complicated multi-***q*** incommensurate magnetic structure in the Neel state. Magnetic ordering is characterized by propagation vector ***q*** = (1/2±δ, 1/2±δ, 1/2±δ) with δ = 0.035 and detected in [53,66,67] for Ho^11^B_12_ in the neutron diffraction experiments at low temperatures in low (*H* < 20 kOe) magnetic field. It was also found for HoB_12_ [53,66,67] that as the strength of external magnetic field increases above 20 kOe, the 4***q***-magnetic structure transforms into a more complex one, in which, apart from the coexistence of two AF 4***q*** and ***2q*** components, there additionally arises some ferromagnetic order parameter. Then, a strong modulation of the diffuse neutron-scattering patterns was observed in HoB_12_ well above *T*_N_ [53,67] with broad peaks at positions of former magnetic reflections, e.g., at (3/2, 3/2, 3/2), pointing to strong correlations between the magnetic moments of Ho^3+^ ions. These diffuse scattering patterns in the paramagnetic state were explained in [53,67] by the appearance of correlated 1D spin chains (short chains of Ho^3 +^ -ion moments placed on space diagonals <111> of the elementary unit), similar to those detected in low dimensional magnets [68]. It was found that these patterns can be resolved both well above (up to 70 K) and below *T*_N_, where the 1D chains seem to condense into an ordered antiferromagnetic modulated (AFM) structure [53,67,69]. The authors [53] discussed the following scenario for the occurrence of long-range order in HoB_12_: Far above *T*_N_, strong interactions lead to correlations along [111], they are essentially one-dimensional and would not lead to long-range order at finite temperature. As *T*_N_ is approached, the 1D-correlated regions grow in the perpendicular directions, possibly due to other interactions. Cigar-shaped AFM-correlated regions were proposed in [53] that become more spherical when *T*_N_ is approached. Within this picture, the ordering temperature is located in the point where spherical symmetry is reached. Only then 3D behavior sets in, and HoB_12_ exhibits long-range AFM order [53]. The refinement of Ho^11^B_12_ crystal structure was done with high accuracy in the space group $Fm\bar{3}m$, but also small static Jahn–Teller distortions were found in RB_12_ compounds [36,41]. However, the most important factor of symmetry breaking is the dynamic one [36,41], which includes the formation both of vibrationally coupled Ho–Ho dimers and dynamic charge stripes (see [36,48,52] for more details). As a result, twofold symmetry in the (110) plane is conserved as expected for cubic crystal, but the charge stripes and Ho–Ho-coupled vibrations suppress the exchange between nearest neighbored Ho-ions. This result in the emergence of complicated phase diagrams in the AF state with a number of different magnetic phases separated by radial and circular boundaries (Maltese Cross type of angular diagrams in RB_12_ [35,42,44,46]). In this scenario AF magnetic fluctuations develop well above T_N_ in HoB_12_ along trigonal axis [111], and dynamic charge stripes along <110> suppress dramatically the RKKY indirect exchange between Ho magnetic moments [44] provoking the formation of cigar-shaped AFM-correlated regions proposed in [53]. In our opinion, these effects are responsible both for the emergence of filamentary structures of fluctuating charges in these nonequilibrium metals and for the formation of spin polarization in the conduction band. This also results in the appearance of a complicated multicomponent Hall effect including two (isotropic and anisotropic) anomalous contributions. 

### 3.2. Mechanisms of AHE in Ho_0.8_Lu_0.2_B_12_

Returning to commonly used classification [51,70], it is necessary to distinguish between the intrinsic and extrinsic AHE. Intrinsic AHE is related to the transverse velocity addition due to Berry phase contribution in systems with strong spin–orbit interaction (SOI), while the extrinsic AHE associated with scattering of charge carriers by impurity centers. However, AHE also arises in noncollinear ferromagnets, in which a nonzero scalar chirality S_i_(S_j_ × S_k_) ≠ 0 leads to the appearance of an effective magnetic field even in the absence of SOI [71], and in magnetic metals with a nontrivial topology of spin structures in real space [72,73,74,75,76]. When interpreting experimental data, a problem of identifying the actual mechanisms of AHE arises [51]. Among the extrinsic AHE, skew scattering, for which the scattering angularly depends on the mutual orientation of the charge carrier spin and the magnetic moment of the impurity, predicts a linear relationship between the anomalous component of the resistivity tensor ρ^an^_H_~ρ_xx_ and usually corresponds to the case of pure metals (ρ_xx_ < 1 µOhm·cm) [77]. In the range of resistances ρ_xx_ = 1–100 µOhm·cm the intrinsic AHE dominates, which is due to the effect of the Berry phase (ρ^an^_H_~ρ_xx_^2^) [78]. The contribution to scattering due to another extrinsic AHE, side jumping [79] with a similar scaling (ρ^an^_H_~ρ^2^_xx0_, where ρ_xx0_ is the residual resistivity of the metal) is usually neglected [51]. In the “dirty limit” (ρ_xx_ > 100 µOhm*cm), an intermediate behavior is observed with a dependence of the form ρ^an^_H_~ρ_xx_*^β^* and with an exponent *β* = 1.6–1.8, which is associated usually with the transition to hopping conductivity [51].

When identifying the AHE mechanism in Ho_0.8_Lu_0.2_B_12_ with a small residual resistivity (ρ_xx0_~1 μOhm·cm, Figure 2a), it is not possible to follow the traditional classification as no presence of the itinerant AHE with an asymptotic ρ^an^_xy_~ρ_xx_^2^, or a skew scattering regime (ρ^an^_H_~ρ_xx_) was found here. However, in order to correctly compare these diagonal and off-diagonal components of the resistivity tensor, it is possible to extract the corresponding anomalous contributions. As can be seen from Figure 2, for **H**||**n**||[111], the anisotropic anomalous component of Hall signal is negligible and, as a result, the reduced Hall resistivity measured in angular experiments in direction **H**||**n** consists of the isotropic contribution only. In addition, according to conclusions made in [43,47], at low temperatures in paramagnetic state the magnetoresistance of Ho_0.8_Lu_0.2_B_12_ consists of isotropic negative and anisotropic positive contributions, the latter being close to zero in the direction **H**||**n**||[111].

In this situation, for estimating the anisotropic components ρ^an^_H_ and ρ^an^_xx_, e.g., for the **n**||[001] sample, it suffices to find the difference ρ^an^_H_(**n**||[001]) = ρ_H_(**n**||[001])-ρ_H_(**n**||[111]) (see Figure 2b) and ρ^an^_xx_(**n**||[001]) = ρ(**n**||[001])-ρ(**n**||[111]) (see Figure 2a). Figure 12 demonstrates the scaling relation between these anisotropic components of ρ^an^_H_ and ρ^an^_xx_ for **H**||[001] and **H**||[110] directions, which leads to the following conclusions: (i) For **H**||[001] a ρ^an^_H_~ρ^an^_xx_^1.7^ dependence is observed over the entire temperature range T ≤ T*~60 K. This regime does not correspond to intrinsic AHE (*β* < 2), while an onset of hopping conductivity (*β* = 1.6–1.8) [51] seems to be an unreal scenario in this good metal (ρ_xx_~1 μOhm·cm, Figure 2a). (ii) On the contrary, for **H**||[110], two anisotropic components of the resistivity tensor appear in the interval T < T_S_~15 K and turn out to be related to each other as ρ^an^_H_~ρ^an^_xx_^0.83^ (Figure 12), which does not favor skew scattering (*β*~1) [77]. Note that the exponent *β* for **H**||[110] is twice as small as that for **H**||[100] in Ho_0.8_Lu_0.2_B_12_, and these regimes are observed in adjacent ρ^an^_xx_ intervals changing one to another at ρ^an^_xx_~0.1 µOhm·cm (Figure 12). Such a different behavior in charge transport parameters for two different magnetic field directions suggests that the AHE is caused by another scattering mechanism, which, in particular, may result from the influence of external magnetic field on dynamic charge stripes directed along <110> (see Figure 1a).

In this scenario the appearance of two types of AHE in Ho_0.8_Lu_0.2_B_12_ may be interpreted as follows. The first mode of AHE associated with charge scattering in the interval T < T_S_~15 K is detected when magnetic field is applied along charge stripes (**H**||[110]), and the regime appears due to the formation of a large size cluster (long chains) in the filamentary structure of fluctuating charges. The second mode of AHE is induced by the order–disorder transition at T*~60 K and corresponds to the magnetic field applied transverse to vibrationally coupled dimers of rare-earth ions (**H**||[001] ⊥ <110>). In the latter case, when the carrier moves in transverse magnetic field along a complex path, the intrinsic AHE is expected to be influenced by the Berry phase in real space [51,78], but instead, the ρ^an^_H_~ρ^an^_xx_^1.7^ scaling is observed. This unusual behavior seems to be a challenge to the contemporary AHE theory and has to be clarified in future studies.

### 3.3. AHE Anisotropy and Dynamic Charge Stripes

The above analysis of Hall effect contributions in Ho_0.8_Lu_0.2_B_12_, based (i) on measurements in the conventional field-sweep ±**H**||**n** scheme (Figure 2, Figure 3a–c and Appendix A), and (ii) on studies of angular dependences with vector **H** rotating in the (110) plane (Figure 5, Figure 6, Appendix A), allows to obtain a set of AHE coefficients, which characterize the ordinary and anomalous contributions along three principal directions of the magnetic field (**H**||**n**||[001], **H**||**n**||[110], and **H**||**n**||[111]).

In this case, the methodological feature of the performed angular measurements of Hall resistivity shows a cosine modulation of the projection of transverse Hall electric field on the direction that connects two Hall probes and is perpendicular to any of the specific normal vectors: **n**||[001], **n**||[110] **n**||[111], or **n**||[211]. In this situation, it is more convenient to control the projection of the external field **H** onto the normal vector H_n_ = (**H**·**n**) = H_0_·cosφ, which is used to determine the amplitude of contributions in the corresponding **n** direction (see inset in Figure 2a). As can be seen from Figure 10, vanishing of the AHE for **H** directed precisely along **n**||[111] and **n**||[112] does not mean zero values of ρ_H_^an^(φ)/H for these crystals in the entire range of angles. In this case, it is obvious that for **H** in the plane of the sample, near-zero values of H_n_ = H_0_·cos occur.

For φ = 0, one should also expect zero values of anomalous contributions to the Hall signal.

In [44,47,48,49,58], it was found that the angular dependence of magnetoresistance (MR) in RB_12_ is determined by scattering of carriers on dynamic charge stripes. As a result, the maximum positive values of MR are observed for **H**||[001] perpendicular to the direction of these electron density fluctuations, while the minimum MR is observed for **H**||[111] (see Figure 13b). To clarify the nature of these anomalies in angular AHE curves, one can restore the angular dependence of the AHE in the entire range of 0–360º and compare the obtained curve with the related MR data. This analysis can be performed by relying on experimental ρ_H_/B(φ) curves measured at T = 2.1 K in magnetic field H = 80 kOe for four different crystals when **H** is rotated in the same plane (110). Since both the ordinary and anomalous components of Hall signal can be described by cosine dependence ρ_H_(φ) = ρ_H0_·cos(φ − φ_1_) + ρ_H_^an^·g(φ), the representation of experimental data shown in Figure 13a in the form of ρ_H_(φ)/(H·cos(φ − φ_1_)) allows us to separate the isotropic and anisotropic contributions from Hall experiments.

The averaged envelope (indicated by yellow shading in Figure 13a) was obtained after removing the particular portions of the related angular dependences with singularities associated with division by small values (zeros of cosine), and then averaging the data of these four angular Hall signal dependences. In this case, in accordance with the data in Figure 2b, in **H**||<111> directions on the resulting envelope curve ρ_H_(φ)/(H·cos(φ − φ_1_)), the maximum negative values of about −6 × 10^−4^ cm^3^/C correspond to isotropic ordinary component ρ_H0_/H of Hall effect, which is independent on magnetic field direction. The positive anisotropic component reconstructed from the data of four measurements (the yellow shading in Figure 13a) provide changes of the ρ_H_(φ)/(H·cos(φ − φ_1_)) in the range (3.2:6)·10^−4^ cm^3^/C. Despite the fact that the initial ρ_H_^an^(φ) dependence is an odd function (see Figure 5, Figure 6, Figure 9 and Figure 10), the result of its division by the odd cos(φ − φ1) allows one to obtain the real anisotropic **even** amplitude of Hall effect and compare it with MR. The location of its extrema coincides with the positions of anomalies on the MR curve (Figure 13b). Indeed, the maximum positive contribution to AHE appears synchronously with the MR peak along <001>, while for <110>, a small (if compared with the anomaly along <001>) positive AHE component is recorded (Figure 13b) simultaneously with a small amplitude singularity of MR. We also note that two spatial diagonals <111> on the anisotropic contribution ρ_H_^an^g(φ)/(H·cos(φ − φ_1_)) seem to be equivalent and show no hysteretic features. The observed behavior of the Hall effect in Ho_0.8_Lu_0.2_B_12_ agrees very well with symmetry lowering of the *fcc* structure of Ho_0.8_Lu_0.2_B_12_ due to static and dynamic Jahn–Teller distortions [36]. 

Finally, the comparison of the angular dependences of MR and Hall effect in Ho_0.8_Lu_0.2_B_12_ (Figure 13) shows that, along with the normal isotropic contributions to the diagonal (negative MR) and off-diagonal (the ordinary Hall coefficient of negative sign) components of the resistivity tensor, anomalous anisotropic positive components appear both in the MR and in Hall effect at low temperatures. These components reach (i) maximal values in the direction of magnetic field transverse to dynamic charge stripes (**H**||[001]) and (ii) zero values for **H**||[111]. This anisotropy arises simultaneously with the transition to the cage glass state at T*~60 K and seems to be related to the formation of vibrationally coupled pairs of rare-earth ions displaced from their centrosymmetric positions in B_24_ cavities of the boron sublattice [48]. A significant increase of this anisotropy is detected at temperatures T < T_S_~15 K upon the formation of large size clusters (long chains) in the filamentary structure of fluctuating electron density (stripes). Taking into account that, according to the results of room temperature measurements of the dynamic conductivity of LuB_12_, about 70% of charge carriers participate in the formation of the collective mode (hot electrons) [80], a redistribution of carriers between the nonequilibrium and Drude components should be expected with decreasing temperature.

Apparently, the activation behavior of the Hall concentration of charge carriers observed in the range 60–300 K (T_a_~14–17 K, Figure 8a) may be attributed to the involvement of additional conduction electrons in the collective mode. When vibrationally coupled dimers of rare-earth ions are formed below T*~60 K, short and disordered chains of stripes oriented along <110> appear in magnetic field, initiating the emergence of intrinsic AHE (Figure 12). We propose that the AHE in Ho_0.8_Lu_0.2_B_12_ is caused by a transverse addition to velocity due to the Berry phase contribution [51], which arises for carriers moving in a complex filamentary structure of the electron density in magnetic field applied transverse to dynamic stripes. During the formation of large size clusters in the structure of stripes (interval T < T_S_~15 K) in field orientation along the stripes **H**||[110], no intrinsic AHE is expected. A skew scattering contribution, for which the scattering angle depends on the mutual orientation of the charge carriers spin and the magnetic moment of the impurity, may become noticeable with a linear relationship ρ^an^_H_~ρ^an^_xx_ between these components of the resistivity tensor (Figure 12). We propose that some geometric factors are responsible for the reduction of *β* exponent in these two AHE regimes. Approaching the AF transition above T_N_, on-site 4*f*-5*d* spin fluctuations in the vicinity of Ho^3+^ ions lead to magnetic polarization of the 5*d* states of the conduction band, which gives rise to ferromagnetic fluctuations in Ho_0.8_Lu_0.2_B_12_ (Figure 4). These produce ferromagnetic nanoscale domains (ferrons), and, as a result, initiate the appearance of the ferromagnetic contribution to AHE (Figure 11 and Table 1). We emphasize that such a complex multicomponent AHE in model Ho_0.8_Lu_0.2_B_12_ metal with a simple *fcc* lattice turns out to be due to the inhomogeneity and complex filamentary structures of the electron density in the matrix of this SCES with dynamic Jahn–Teller lattice instability and electron phase separation.

## 4. Experimental Details

Ho_0.8_Lu_0.2_B_12_ single-domain crystals were grown by crucible-free induction zone melting in an inert argon gas atmosphere (see, e.g., [34]). Magnetization was measured with the help of a SQUID magnetometer MPMS Quantum Design in fields up to 70 kOe in the temperature range 1.9–10 K. The external magnetic field was applied along the principal crystallographic axes- **H**||[001], **H**||[110], and **H**||[111]. Resistivity, magnetoresistance, and Hall effect were studied on an original setup using the standard DC five-probe technique with excitation current commutation. The angular dependences of transverse magnetoresistance and Hall resistivity were obtained using a sample holder of original design, which enables the rotation of the vector **H** located in the plane perpendicular to fixed current direction **I**||[$1\bar{1}0$]⊥**H** with a minimum step φ_step_ = 0.4° (see the schematic view on the inset of Figure 2a). Measurements were carried out in a wide temperature range 1.9–300 K in magnetic fields up to 80 kOe, the angle φ = **n**^**H** (**n**-normal vector to the lateral sample surface) varied in the range φ = 0–360°. The measuring setup was equipped with a stepper motor, which enabled an automatic control of sample rotation, similar to that one used in [35]. High accuracy of temperature control (ΔT ≈ 0.002 K in the range 1.9–7 K) and magnetic field stabilization (ΔH ≈ 2 Oe) was ensured, respectively, by LLC Cryotel, (Moscow, Russia) TC 1.5/300 temperature controller and Cryotel SMPS 100 superconducting magnet power supply in combination with a CERNOX 1050 thermometer and n-InSb Hall sensors.

## 5. Conclusions

In the paramagnetic phase of the cage-cluster high-boron boride Ho_0.8_Lu_0.2_B_12_ with a cage-glass state below T*~60 K and electronic phase separation (dynamic charge stripes), magnetotransport was studied at temperatures 1.9–300 K in magnetic fields up to 80 kOe. Field and angular measurements of resistivity and Hall resistivity were performed on single-domain crystals of the Ho_0.8_Lu_0.2_B_12_ model metal allowed to separate and analyze several different contributions to the Hall effect. It was shown that, along with the negative ordinary isotropic component of Hall resistivity, an intrinsic AHE of a positive sign arises in the cage-glass state in field direction **H**||[001], which is perpendicular to the charge stripe chains. This AHE corresponds through the relation ρ^an^_H_~ρ^an^_xx_^1.7^ to the anomalous components of the resistivity tensor. It was also found that at temperatures T < T_S_~15 K, where long chains prevail in the filamentary structure of fluctuating charges (stripes), a contribution to AHE of the form ρ^an^_H_~ρ^an^_xx_^0.83^ becomes dominant when **H**||[110]. We propose that these two components are intrinsic (a transverse addition to velocity due to the contribution of the Berry phase) and extrinsic (from the skew scattering mechanism) [51], respectively, and exhibit some decrease of exponents from integers due to the geometric factor. In the paramagnetic phase near Neel temperature, on-site 4*f*-5*d* spin fluctuations in the vicinity of Ho^3+^ ions were found to induce spin-polarized 5*d* states (ferromagnetic nanoscale domains–ferrons) in the conduction band, which result in the appearance of an additional ferromagnetic contribution to AHE, as observed both in the isotropic and anisotropic components of Hall effect. Detailed measurements of the angular dependences of Hall resistivity and MR with vector **H** rotation in the (110) plane, perpendicular to the direction of stripes, made it possible to separate the negative isotropic and positive anisotropic contributions to AHE and MR, and to explain them in terms of charge carriers scattering by dynamic charge stripes.

## Figures and Tables

**Figure 1 molecules-28-00676-f001:**
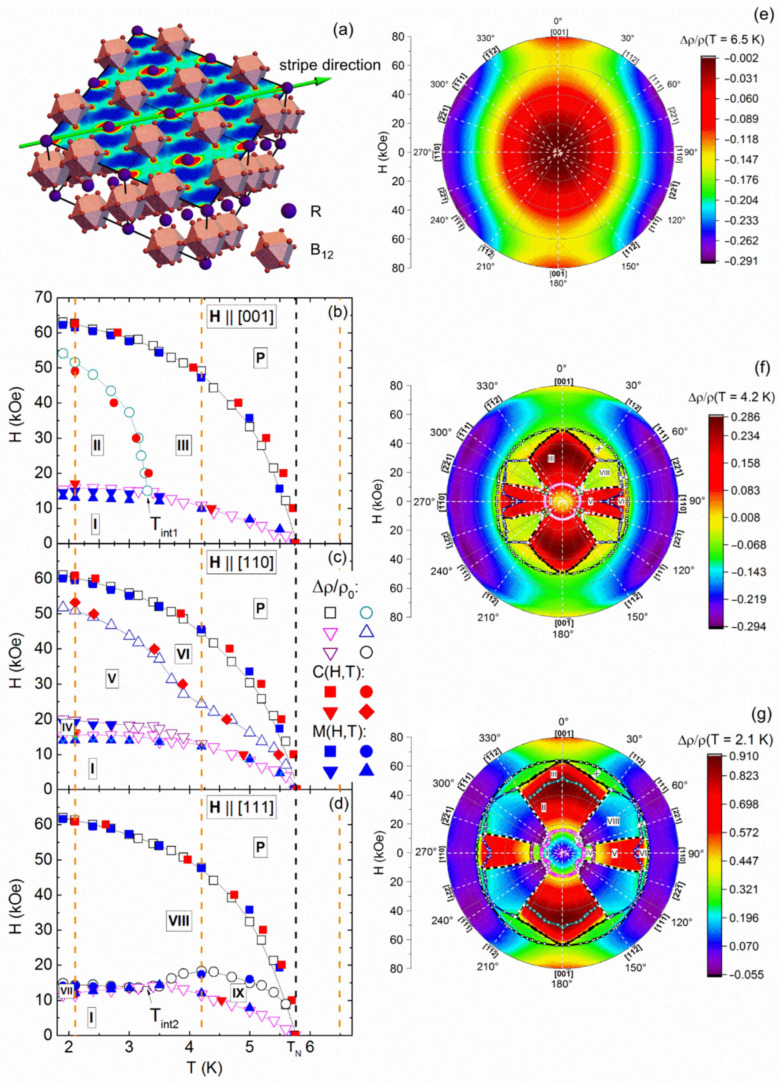
(**a**) Sketch of charge stripes arrangement (green lines) in the RB_12_ crystal structure, (**b**–**d**) H-T magnetic phase diagrams of the AF state of Ho_0.8_Lu_0.2_B_12_ in different field directions, (**e**) polar plot of the field-angular magnetoresistance dependence in the paramagnetic state and (**f**,**g**) angular H-φ magnetic phase diagrams (color shows the magnetoresistance amplitude) of the AF state of Ho_0.8_Lu_0.2_B_12_ at low temperatures (reproduced from [35]).

**Figure 2 molecules-28-00676-f002:**
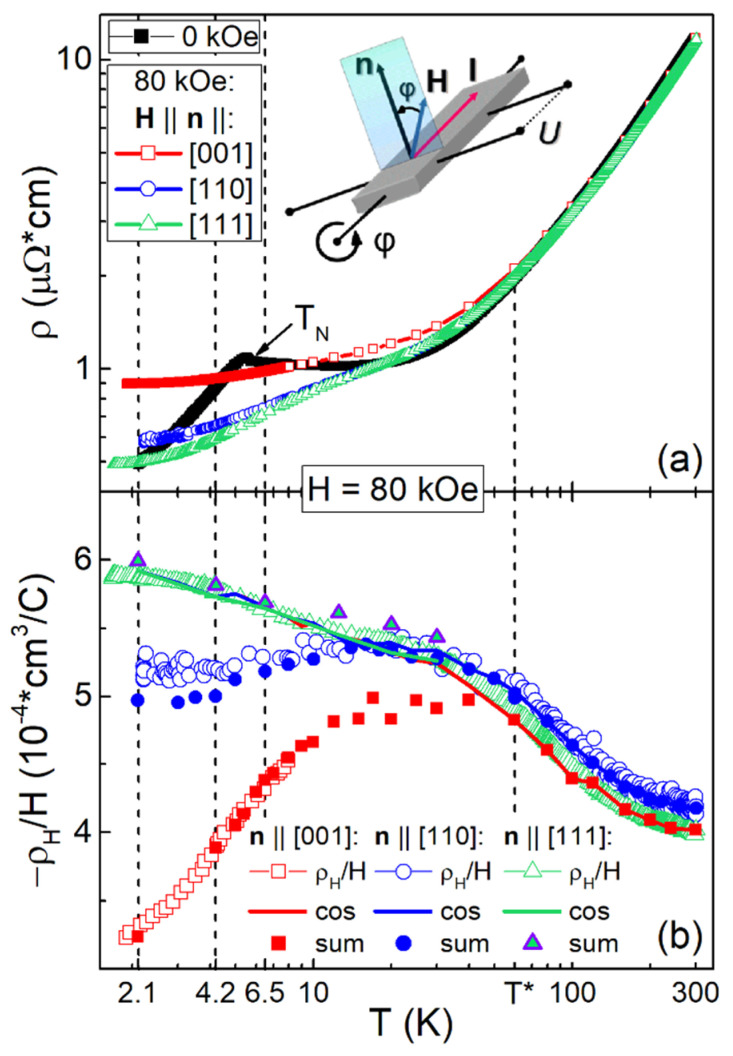
Temperature dependences (**a**) of resistivity ρ(T) at H = 0 and H = 80 kOe, as well as (**b**) the absolute value of reduced Hall resistivity –ρ_H_(T)/H in Ho_0.8_Lu_0.2_B_12_ for samples with **H**||**n**||[001], **H**||**n**||[110], and **H**||**n**||[111] (see inset). On panel (**b**) open and closed symbols show the experimental data for -ρ_H_/H and the sum of isotropic and anisotropic contributions to Hall effect sum = ρ_H0_/H + ρ_H_^an^/H, correspondingly. Thick solid lines show the reduced amplitudes of the isotropic component cos = ρ_H0_/H (see Section 2.3 for details). Vertical dashed lines point to the transition to the cage-glass state at T*~60 K [37] and to the formation of stripes at T_S_~15 K (see discussion below), and denote the temperatures 2.1 K, 4.2 K, and 6.5 K at which Hall resistivity was studied in more detail.

**Figure 3 molecules-28-00676-f003:**
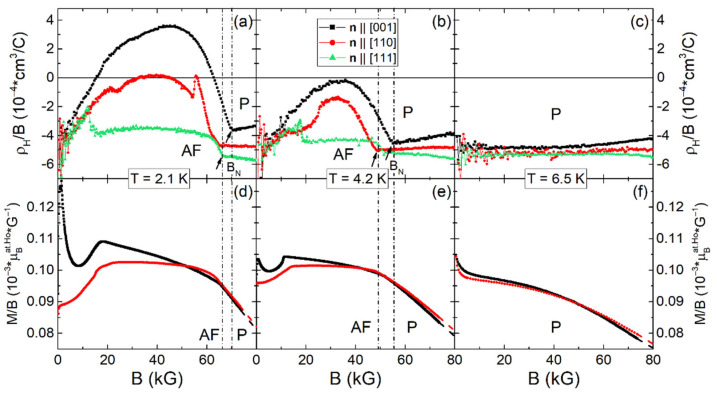
(**a**–**c**) Field dependences of the reduced Hall resistivity ρ_H_/B vs magnetic induction **B** at T = 2.1, 4.2, and 6.5 K for samples with **H**||**n**||[001], **n**||[110], and **n**||[111], respectively. Arrows at **B**_N_ indicate the AF-P transitions. (**d**–**f**) Corresponding curves of magnetic susceptibility M/B(B). Dashed lines show the approximation in the interval 7–8 T (see text).

**Figure 4 molecules-28-00676-f004:**
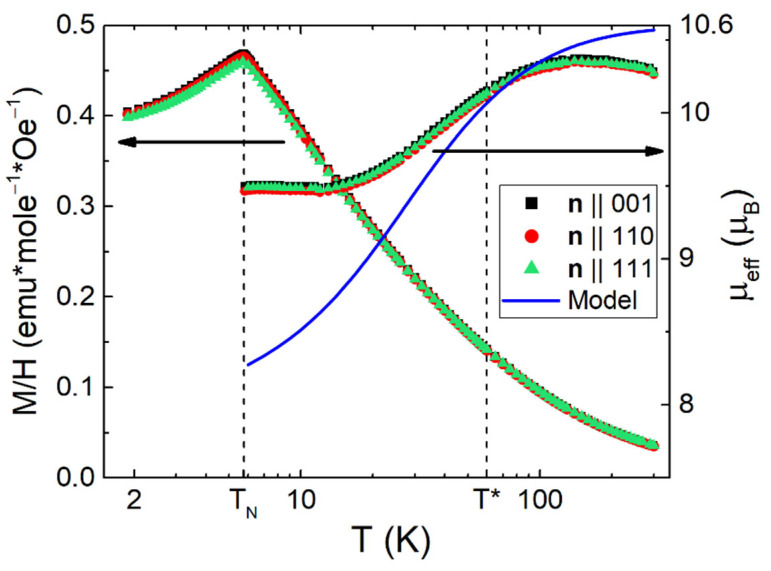
Temperature dependences of magnetic susceptibility (left scale) and of the effective moment (right scale) at H = 100 Oe for different samples with **H**||**n**||[001], **n**||[110], and **n**||[111] (see text). Solid line indicates the changes of the magnetic moment of Ho ^5^*I*_8_-multiplet splitting by CEF in HoB_12_ (see [52]).

**Figure 5 molecules-28-00676-f005:**
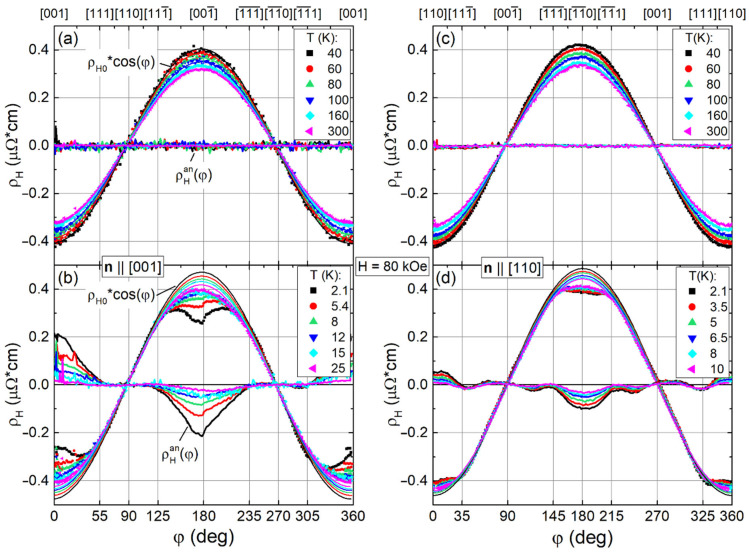
(**a**–**d**) Angular dependencies of Hall resistivity ρ_H_(φ) measured in H = 80 kOe for samples with **n**||[001] and **n**||[110] in the temperature range 2–300 K. Symbols show the experimental ρ_H_(φ) data, thin and thick curves demonstrate the isotropic f_cos_(φ) ≈ ρ_H0_·cos(φ) and anisotropic ρ_H_^an^(φ) = ρ_H_(φ) − f_cos_(φ) contributions, correspondingly.

**Figure 6 molecules-28-00676-f006:**
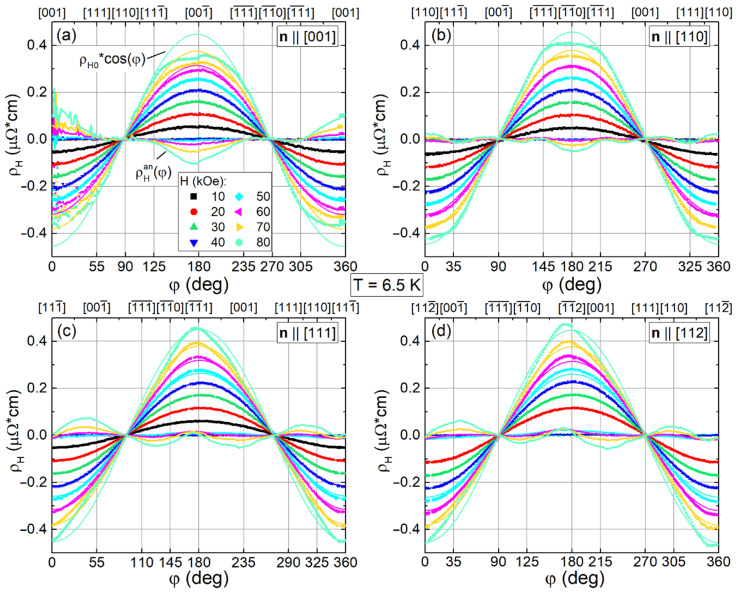
(**a**–**d**) Angular dependences of Hall resistivity ρ_H_(φ) at T = 6.5 K in fixed magnetic field up to 80 kOe for samples with **n**||[001], **n**||[110], **n**||[111], and **n**||[112]. Symbols show the experimental ρ_H_(φ) curves, thin and thick lines indicate isotropic f_cos_(φ) ≈ ρ_H0_·cos(φ) and anisotropic ρ_H_^an^(φ) = ρ_H_(φ) − f_cos_(φ) contributions, correspondingly.

**Figure 7 molecules-28-00676-f007:**
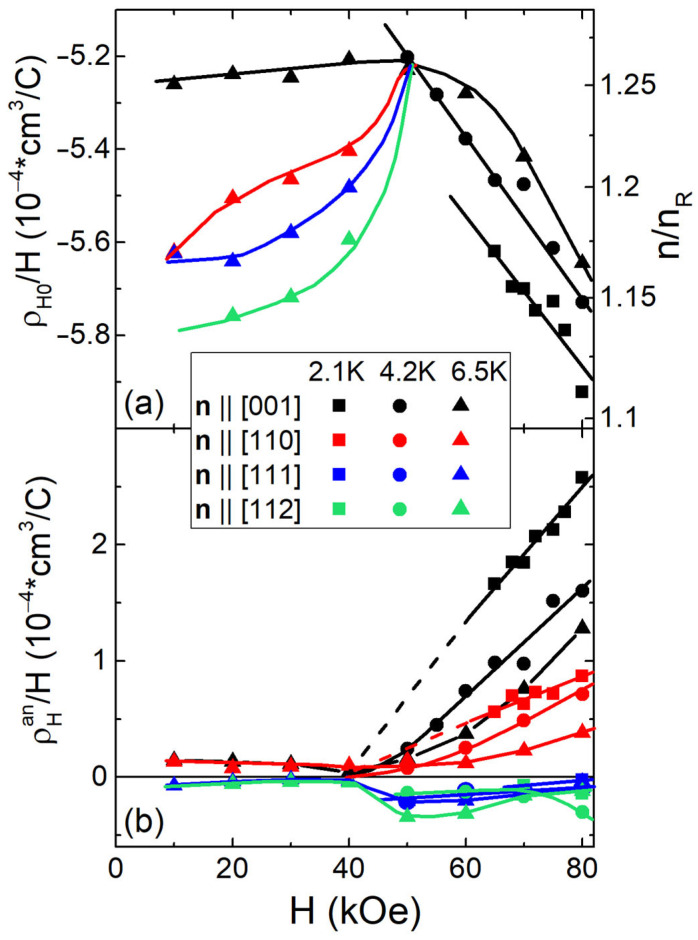
Reduced amplitudes of (**a**) isotropic ρ_H0_/H and (**b**) anisotropic ρ_H_^an^/H contributions vs external magnetic field H at temperatures of 2.1, 4.2, and 6.5 K for samples with **n**||[001], **n**||[110], **n**||[111], and **n**||[112]. Different temperatures are indicated by different shapes of symbols, while samples with different **n** directions are indicated by different colors of the symbols. Right axis on panel (**a**) shows the reduced Hall concentration n/n_R_ for comparison (see text).

**Figure 8 molecules-28-00676-f008:**
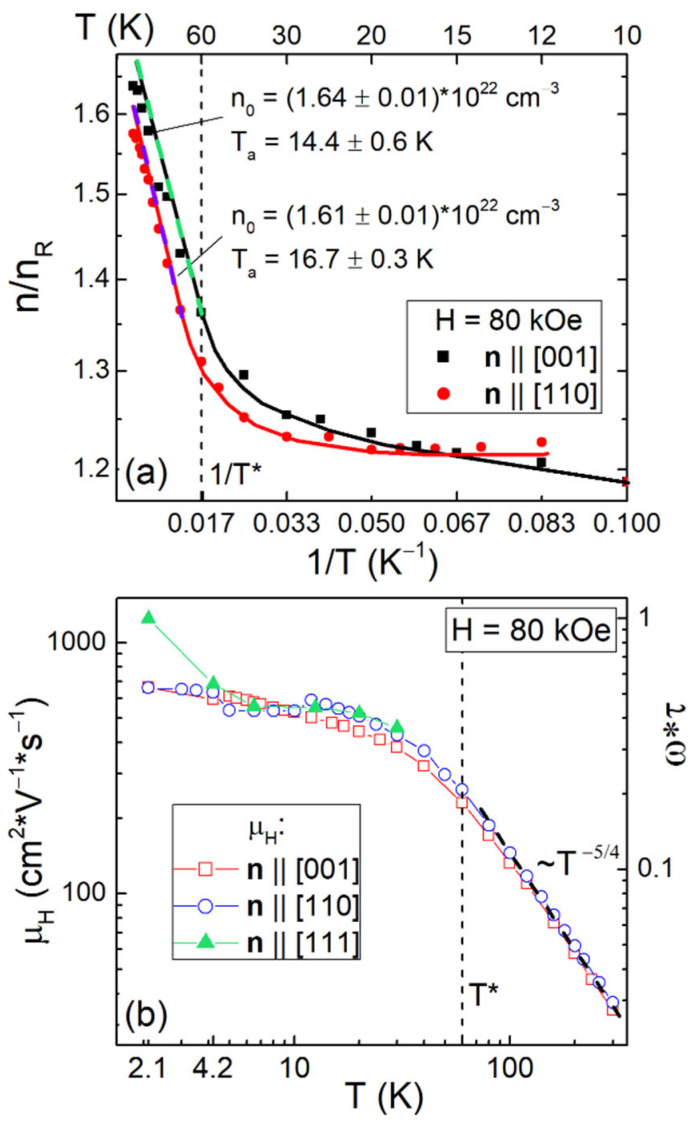
(**a**) Arrhenius plot *lg(n/n_R_) = f(1/T)* of the reduced Hall concentration for **n**||[001] and **n**||[110] at H = 80 kOe. (**b**) Temperature dependences of the Hall mobility *µ*_H_ and the parameter *ω_c_τ ≈ µ*_H_*·H* for three directions **n**||[001], **n**||[110], and **n**|[111]. Thick dashed lines show the (**a**) activation behavior and (**b**) power law.

**Figure 9 molecules-28-00676-f009:**
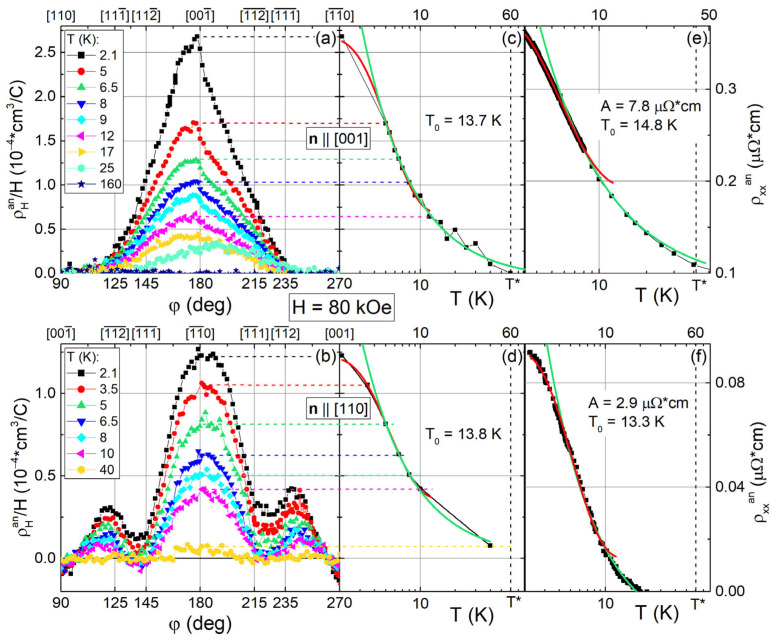
(**a**,**b**) Anisotropic positive contribution ρ_H_^an^(φ)/H for samples with **n**||[001] and **n**||[110] at H = 80 kOe. (**c**,**d**) Temperature dependencies of ρ_H_^an^/H amplitudes for **n**||[001] and **n**||[110] in the logarithmic scale. Panels (**e**,**f**) show the temperature dependencies of the anisotropic contribution to resistivity ρ_xx_^an^ = ρ(**n**||[001], T_0_, H = 80 kOe)-ρ(**n**||[111], T_0_, H = 80 kOe) and ρ_xx_^an^= ρ(**n**||[110], T_0_, H = 80 kOe) − ρ(**n**||[111], T_0_, H = 80 kOe), respectively. Green and red solid lines on panels (**c**–**f**) show the approximation by Equations (3) and (4) (see text).

**Figure 10 molecules-28-00676-f010:**
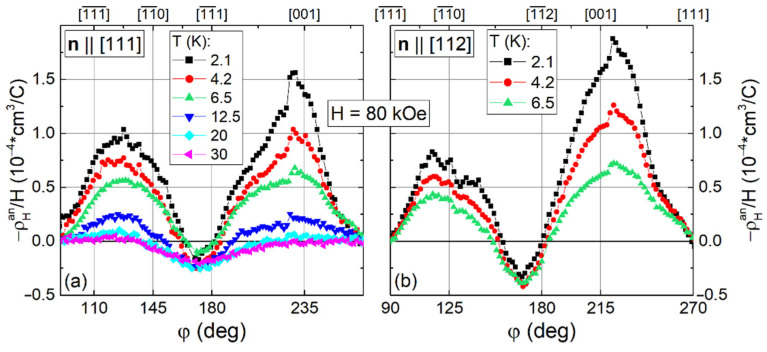
(**a**,**b**) Anisotropic contributions ρ_H_^an^(φ)/H in magnetic field H = 80 kOe for the samples with **n**||[111] and **n**||[112], respectively.

**Figure 11 molecules-28-00676-f011:**
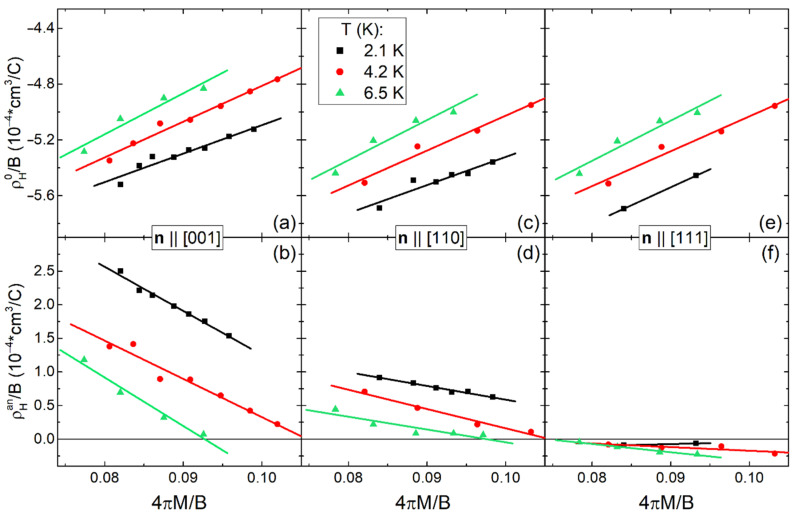
Linear approximation of the isotropic ρ_H0_/B (**a**,**c**,**e**) and anisotropic ρ_H_^an^/B (**b**,**d**,**f**) contributions vs 4πM/B within the Equation (6) approximation at 2.1–7 K for three principal directions **H**||[001], **H**||[110], and **H**||[111] (see text).

**Figure 12 molecules-28-00676-f012:**
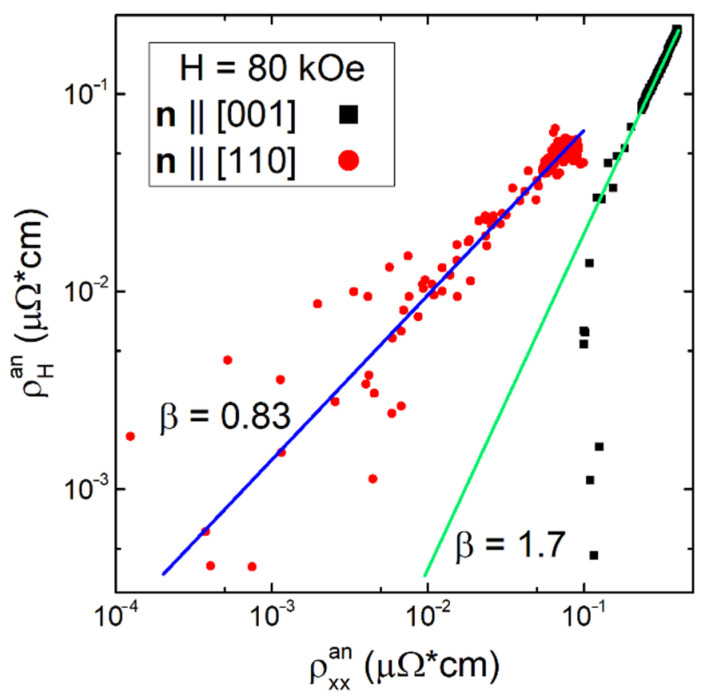
Anisotropic AHE components for directions **H**||[001] and **H**||[110] in magnetic field H = 80 kOe scaled in double logarithmic plot. Solid lines display the linear approximations and *β* denotes the exponent in ρ^an^_H_~ρ^an^_xx_*^β^*.

**Figure 13 molecules-28-00676-f013:**
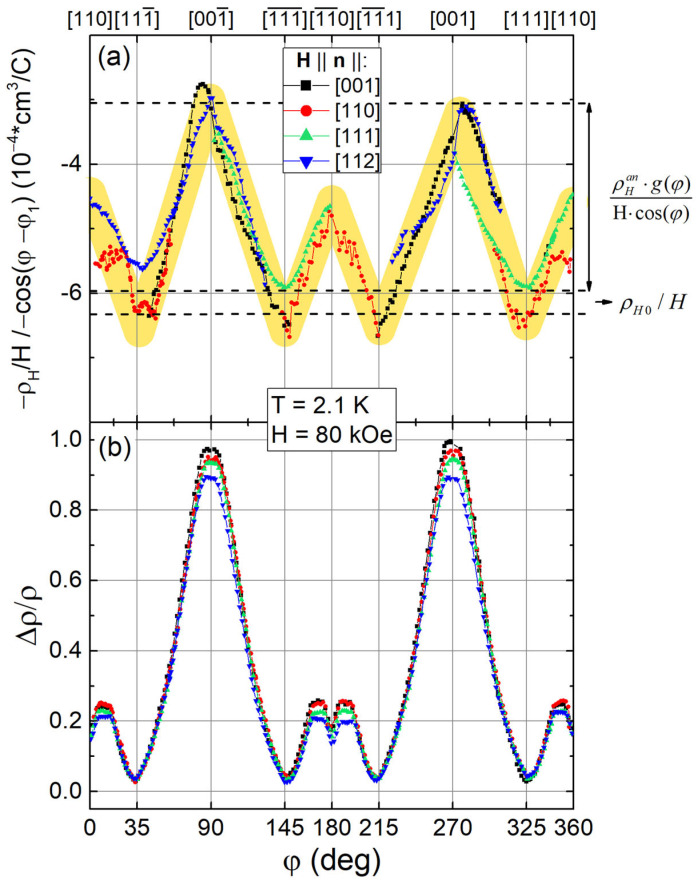
(**a**) Normalized angular Hall resistivity ρ_H_(φ)/(H·cos(φ − φ_1_)) and (**b**) magnetoresistance curves in field 80 kOe at temperature 2.1 K for four samples with **n**||[001], **n**||[110], **n**||[111], and **n**||[112]. Yellow shading indicates the common envelope for all four Hall effect measurements (see text).

**Table 1 molecules-28-00676-t001:** Parameters R_H_^0^, R_H_^an^ of the ordinary and anomalous R_M_^0^, R_M_^an^ contributions to Hall effect in Ho_0.8_Lu_0.2_B_12_ averaged over temperatures 2.1–7 K.

R_H_, 10^−4^ × cm^3^/C	H||n||[001]	H||n||[110]	H||n||[111]
R_H_^0^	−7.3	−7.7	−7.7
R_H_^an^	6.8	2.3	0
R_M_^0^	25.2	26.9	26.5
R_M_^an^	−64.9	−20.9	0

## Data Availability

The data presented in this study are available on request from the corresponding author.

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
