# Peer review of "Hall Effect Anisotropy in the Paramagnetic Phase of Ho0.8Lu0.2B12 Induced by Dynamic Charge Stripes"

_molecules, 2023, doi:10.3390/molecules28020676_

Round 1
Reviewer 1 Report
The manuscript is well written and the charge transport measurements are very detailed. this is a good paper a continuation of the authors study into the LuB12 and Lu0.5Ho0.5B12 systems. The authors do a great job of explaining each measurement in good detail. The manuscript is appropriate for this journal but will need a few corrections.
1) Can the authors add a powder XRD data to confirm the phase purity.
2) For the crystal in any one of the directions say the [111] can the authors provide that information using XRD or TEM or electron diffraction pattern.
3) For the composition confirmation can the authors provide EDX to confirm a homogeneous composition across the ingot.
4) The authors do a great job of explaining the intrinsic contribution to the AHE but dont talk a lot about the extrinsic contributions. Can the authors elaborate on that in the main text
5) There are many grammatical errors within the manuscript which must be addressed before the manuscript can be accepted.
Author Response
1) Can the authors add a powder XRD data to confirm the phase purity.
2) For the crystal in any one of the directions say the [111] can the authors provide that information using XRD or TEM or electron diffraction pattern.
3) For the composition confirmation can the authors provide EDX to confirm a homogeneous composition across the ingot.
Our reply.
In accordance with the Referee recommendation we include in the Supplementary Information (see SI, Section 3. Crystal orientation and quality control):
1) the powder XRD for Ho0.8Lu0.2B12 (Fig. S5),
2) X-ray Laue backscattering patterns for Ho0.8Lu0.2B12 crystals with directions [100], [110] and [111] (Figs. S6a-c) and
3) results of the microprobe analysis in single crystal with nominal composition Ho0.8Lu0.2B12 (a TEM general view of the lateral cross section of an as-grown crystal, Fig.10 and Table S2) to confirm the homogeneous composition across the ingot. The real / exact composition of the crystal is Ho0.809±0.004Lu0.191±0.004B12.
Moreover, the rocking curve control for a single-domain crystal was also applied (see Section 3.3 and Figs. S7-S9 in the Supplementary Information, measurements were performed on Bruker D8 Discover x-ray diffractometer).
4) The authors do a great job of explaining the intrinsic contribution to the AHE but dont talk a lot about the extrinsic contributions. Can the authors elaborate on that in the main text
Our reply.
In order to avoid misleading of the Reader we included in the main text the clarification that both skew scattering and side-jumping are impurity induced extrinsic contributions to the AHE.
5) There are many grammatical errors within the manuscript which must be addressed before the manuscript can be accepted.
Our reply.
The manuscript was carefully checked once more and the grammatical errors corrected.
Reviewer 2 Report
In this paper, the author focused on the charge transport in the paramagnetic phae of Ho0.8Lu0.2B12 crystals and tried to explore the possible intrinsic mechanism. This work is interesting, which can shed light on the future study of strongly correlated electron systems. But there are some questions needed to be answnered by the authors.
1. The importance of this work needs to be further emphasized, including the possible application of the Ho0.8Lu0.2B12 crystals.
2. There are many complex or insuitable sentences in this paper, such as "with temperature dependent .eff(T) indicatessuch as..."(Page 8) ; "As the population of excited magnetic states of the Ho3+ 5I8 multiplet splitting by crystal electric field (CEF)..." (Page 8), "below 40 K the ρHan(φ) curves for n||[001] exhibit a broad feature in a wide range of angles around <001> and between the<001> and <111> axes... (Page 9), "It is worth noting that in the range T>T*~60 K the temperature dependences of reduced Hall resistivity ρH/H(T) at H=80 kOe for samples with n||[001], n||[110]...(Page 10)"So I suggest the authors should smooth this paper very carefully.
3. What is the crystalline structure and growth direction of the as-grown samples? Can the lattice structure and crystallinity take effect on their Hall resistivity and AHE effect?
4. Why does the author chose the REB12 with fcc phase as the research goal rather than REB6 (SmB6, CeB6, etc)? What is the advantages?
5. What is the effect of the Lu contents on the Hall resistivity or magnetoresistance of HoB12 crystals?
6. What are the Hall resistivity and magnetoresistance performances of the Ho0.8Lu0.2B12 crystal in comparison with the reported results?
Author Response
1). The importance of this work needs to be further emphasized, including the possible application of the Ho0.8Lu0.2B12 crystals.
Our reply.
We added the sentence
“RB12 (R-Tb, Dy, Ho, Er, Tm, Yb and Lu) compounds attract considerable attention for practical applications due to the unique combination of physical properties, such as high melting points, microhardness and high chemical resistance, which create prospects for their practical applications [30-33]. These materials are extremely interesting also for fundamental studies.”
in the introduction to satisfy the recommendation of Referee B.
2). There are many complex or insuitable sentences in this paper, such as "with temperature dependent .eff(T) indicates such as..."(Page 8) ; "As the population of excited magnetic states of the Ho3+ 5I8 multiplet splitting by crystal electric field (CEF)..." (Page 8), "below 40 K the ρHan(φ) curves for n||[001] exhibit a broad feature in a wide range of angles around <001> and between the<001> and <111> axes... (Page 9), "It is worth noting that in the range T>T*~60 K the temperature dependences of reduced Hall resistivity ρH/H(T) at H=80 kOe for samples with n||[001], n||[110]...(Page 10) "So I suggest the authors should smooth this paper very carefully.
Our reply.
The sentences were corrected and improved.
3). What is the crystalline structure and growth direction of the as-grown samples? Can the lattice structure and crystallinity take effect on their Hall resistivity and AHE effect?
Our reply.
All single crystals of the HoxLu1-xB12 system were grown with seeds oriented along [001], [110] and [111] directions. Deviations from the given orientation in as-grown crystals were eliminated during sample preparation. Special attention was paid to choose only single-domain crystals (without any twinning or / and multi-domain crystallinity) for the study of Hall effect anisotropy (for more details see Section 3 in SI).
4). Why does the author chose the REB12 with fcc phase as the research goal rather than REB6 (SmB6, CeB6, etc)? What is the advantages?
Our reply.
For the first time a study of Hall effect anisotropy in metal with dynamic charge stripes was carried out on single-domain crystals of LuB12 (see Ref. 48 in the revised manuscript). The study presented here is addressed to more a complicated dodecaboride Ho0.8Lu0.2B12, which is a metal with magnetic Ho3+ ions. In future we are planning to examine REB6 compounds (e.g. SmB6, CeB6, etc.) to look for the dynamic charge stripes, and after that a study of AHE will be undertaken.
5)-6). What is the effect of the Lu contents on the Hall resistivity or magnetoresistance of HoB12 crystals? What are the Hall resistivity and magnetoresistance performances of the Ho0.8Lu0.2B12 crystal in comparison with the reported results?
Our reply.
The substitution of Lu3+ by magnetic Ho3+ ions induces the emergence and intensification of magnetic scattering in HoxLu1-xB12, resulting to a decrease of both the Hall and drift mobility of the charge carriers. The crystals Ho0.8Lu0.2B12, which are the best single-domain crystals in the family HoLuB12 compounds were taken to investigate in details both the Hall effect and magnetoresistance contributions and their anisotropy, which presents the first complex study of resistivity tensor components in metallic high-boron borides with magnetic ions.
Round 2
Reviewer 1 Report
The authors have responded to all the comments. Accept as is.